# Community based reference interval of selected clinical chemistry parameters among apparently healthy Adolescents in Mekelle City, Tigrai, Northern Ethiopia

**Getachew Belay**[1]*, **Gebreyohanes Teklehaymanot**[2], **Gebreslassie Gebremariam**[3], **Kelali Kaleaye**[4], **Hagos Haileslasie**[5], **Gebremedhin Gebremichail**[5], **Brhane Tesfanchal**[1], **Getachew Kahsu**[1], **Brhane Berhe**[6], **Kebede Tesfay**[6], **Lemlem Legesse**[4], **Ataklti Gebretsadik**[4], **Mistire Wolde**[7], **Aster Tsegaye**[7]

1 Unit of Clinical Chemistry, Department of Medical Laboratory Sciences, College of Medicine and Health Sciences, Adigrat University, Adigrat, Ethiopia, 2 Unit of Hematology and Immuno-Hematology, Department of Medical Laboratory Sciences, College of Medicine and Health Sciences, Mekelle University, Mekelle, Ethiopia, 3 Unit of Clinical Chemistry, Department of Medical Laboratory Sciences, College of Medicine and Health Sciences, Mekelle University, Mekelle, Ethiopia, 4 Laboratory Diagnostic, Research and Quality Assurance Directorate, Tigrai Health Research Institute, Mekelle, Ethiopia, 5 Unit of Hematology and Immuno-Hematology, Department of Medical Laboratory Sciences, College of Medicine and Health Sciences, Adigrat University, Adigrat, Ethiopia, 6 Unit of Medical Parasitology and Entomology, Department of Medical Laboratory Science, College of Medcine and Health Science, Adigrat University, Adigrat, Ethiopia, 7 Department of Medical Laboratory Sciences, College of Health Sciences, Addis Ababa University, Addis Ababa, Ethiopia

* getabelay65@gmail.com

## Abstract

### Background

Locally established clinical laboratory reference intervals (RIs) are required to interpret laboratory test results for screening, diagnosis and prognosis. The objective of this study was establishing reference interval of clinical chemistry parameters among apparently healthy adolescents aged between 12 and 17 years in Mekelle, Tigrai, northern part of Ethiopia.

### Methods

Community based cross sectional study was employed from December 2018 to March 2019 in Mekelle city among 172 males and 172 females based on Multi stage sampling technique. Blood samples were tested for Fasting blood sugar (FBS), alanine aminino transferase (ALT), aspartate amino transferase (AST), alkaline phosphatase (ALP), Creatinine, urea, total protein, albumin (ALB), direct and indirect bilirubin (BIL.D and BIL.T) using 25 Bio system clinical chemistry analyzer. Results were analyzed using SPSS version 23 software and based on the Clinical Laboratory Standard Institute (CLSI)/ International Federation of Clinical Chemistry (IFCC) C 28-A3 Guideline which defines the reference interval as the 95% central range of 2.5th and 97.5th percentiles. Mann Whitney U test, descriptive statistics and box and whisker were statistical tools used for analysis.

**Data Availability Statement:** All relevant data are within the manuscript.

**Funding:** The author(s) received no specific funding for this work.

**Competing interests:** The authors have declared that no competing interests exist.

## Results

This study observed statistically significant differences between males and females in ALP, ALT, AST, Urea and Creatinine Reference intervals. The established reference intervals for males and females, respectively, were: ALP (U/L) 79.48–492.12 versus 63.56–253.34, ALT (U/L) 4.54–23.69 versus 5.1–20.03, AST 15.7–39.1 versus 13.3–28.5, Urea (mg/dL) 9.33–24.99 versus 7.43–23.11, and Creatinine (mg/dL) 0.393–0.957 versus 0.301–0.846. The combined RIs for Total Protein (g/dL) was 6.08–7.85, ALB (g/dL) 4.42–5.46, FBS(mg/dL) 65–110, BIL.D (mg/dL) 0.033–0.532, and BIL.T (mg/dL) 0.106–0.812.

## Conclusions

The result showed marked difference among sex and with the company derived values for selected clinical chemistry parameters. Thus, use of age and sex specific locally established reference intervals for clinical chemistry parameters is recommended.

## Introduction

Since the day of Gräsbeck and Fellman who introduced the concept of 'Normal Values and Statistics' and the subsequent launching of a new concept of reference values in 1969 by Gräsbeck and Saris, the terminology come in to common practice[1, 2]. From1987 to 1991, the International Federation of Clinical Chemistry (IFCC) published a series of 6 papers, in which it was recommended that each laboratory follow defined procedures to produce its own reference interval[3]. Although there were very important developments and implementations between the 1990s and 2008, the C28A3 guideline, published in 2008 by Clinical Laboratory Standard Institute (CLSI) and IFCC constituted the most significant step in the development of RI and is still in current use [3, 4].

When a test is used for disease screening, diagnosis or prognosis, the test result is normally compared with a normal range that is defined as reference value for a healthy population [5]. The few studies that have focused on reference intervals for adolescents in Africa reported significant differences between adults and adolescents thus indicating the need to have age specific reference intervals when reporting laboratory results for informed decisions to be made [6].

Reference interval (RI) is the interval between, and including, two reference limits used to separates healthy from diseased individuals [7]. International guidelines recommend each clinical laboratory and diagnostic test manufacturers to establish their own RIs belonging to a group of homogenous population. However, the majority of clinical laboratories in the world adopt RIs established by manufacturers [7, 8].

Establishing reference intervals has always been a challenge as significant differences may exist in disease frequencies, biological variation in analytes among ethnic groups, genders and ages as well as specimen collection techniques and test performance [9, 10].

Additional issues and challenges come into play when establishing reference intervals for use in children and adolescents. Differences in physical size, organ maturity, body fluid compartments, (rates of) growth and development, immune and hormone responsiveness, nutrition and metabolism are among the many factors that can influence normal analyte levels in children and adolescents. Also, there are diseases that children are more susceptible, in acquiring than adults. Furthermore, separate reference intervals may be necessary for children of different age groups and/or genders, as well as for neonates, and premature babies [11].

While sexual characteristic changes across puberty are profound, the earlier changes during the growth of a child, from birth to puberty are also significant. The dominant form of partitioning applied in clinical laboratory medicine is by social consensus. Partitioning in adolescence should ideally be linked to the pubertal tanner stages [12].

The results of this study, therefore, will be used as reference values in the future evidence informed practices particularly for adolescents in Mekelle town. Patients will get better service as their result will be interpreted based on the locally established value; physicians will have better tool in their patient management process and medical laboratory professionals will have confidence especially flagging of results generated by automated clinical chemistry analyzers based on RIs established elsewhere are a common challenge.

## Materials and methods

### Study design and period

A cross sectional community based study was employed from December 2018 to March 2019 in Mekelle city, Tigrai National Regional state, North Ethiopia.

### Study area and population

Mekelle is the capital city of Tigrai National Regional state. It is located around 780 kilometers (480 mi) north of the Ethiopian capital Addis Ababa, with an elevation of 2,254 meters (7,395 ft) above sea level and in a semi-arid area with a mean annual rainfall of 714 millimetres (28.1 in). The total population of Mekelle city is 310,436 according to 2007 census. Administratively, Mekelle is considered a Special Zone, which is divided into seven sub-cities. The seven sub cities of Mekelle are Hawelti, Adi-Haki, Kedamay Weyane, Hadnet, Ayder, Semien and Quiha. Within each local administration, there are Kebeles or Ketenas (smaller administration units). Our study was conducted in three randomly selected sub cities, namely, Hawelti, Semen and Ayder.

### Sample size and sampling techniques

For establishing reference interval the Clinical Laboratory Standards Institute (CLSI) guideline which was developed through consensus process for the global application was employed. CLSI recommends that the best means to establish a reference interval is to collect samples from a sufficient number of reference individuals to yield a minimum of 120 samples for analysis, by non-parametric means for each partition (e.g. sex, age range) with a power of 90% [4]. In the current study, the maximum partition needed was 12–17 years old female and male. This age partition is based on previous studies conducted in Canada for AST and ALT [6, 13] and a study revealing pubertal change of serum creatinine level [12]. Thus, two partition groups were needed (2 x 120 = 240).

According to previous studies in other African countries, in such studies about 30% of apparently healthy population [14] do not qualify for reference interval determination for various reasons when tested for the common viral infections and syphilis. Considering a 30% exclusion from data analysis, to reach the CLSI recommended total sample size of 240 for the reference interval determination, a total of 344 individuals were enrolled (i.e, 30% x 344 = 104 to be excluded during data analysis.

Thus, 344 participants were recruited from Mekelle city. The study participants were selected using systematic sampling technique by considering sub-cities as a sampling frame and then households the final selection units. One individual in the household fulfilling the eligibility criteria and willing to participate was included in the study.

**Inclusion criteria.** Young apparently healthy children who lived in Mekelle city at least for 5 years and willing to participate were included in the study.

### Exclusion criteria

- Individuals with known chronic illnesses like diabetes mellitus, chronic renal insufficiency, hypertension, ischemic heart disease, anemia, thyroid, liver diseases, and cancer of any type.

- Individuals who had known infectious disease (HIV, Hepatitis)

- Individuals taking pharmacologically active substances and all prescription drugs.

- Individuals who had Hemo-parasite and intestinal parasite.

- Individuals who received blood transfusion within the previous 1 year.

- Pregnant females were excluded from the study.

### Operational definitions

**Reference interval:** is the interval between, and including, two reference limits (2.5 th and 97.5th percentile) for apparently healthy individuals.

**Adolescent**: young children who are in a markedly pubertal change aged between 12–17 years.

**Selected clinical chemistry tests:** in this study refer to the following biochemical analytes: FBS, ALP, ALT, AST, BIL. D, BIL. T, Urea, Creatinine, Total protein and Albumin.

### Study procedure and data collection

**Data collection and laboratory analysis.** Data collectors were trained for three days about the objective of the study, study participants' rights, confidentiality of information, procedure of physical examination, procedure of blood sample collection and measurements, and how to approach and interview participants before the actual data collection. The study participants were invited to come to the nearby health institution. Study participants who agreed to give written consent after being informed about the purpose of the study and associated risks were physically examined and interviewed. Socio-demographic data and biological samples (urine, stool and blood) were collected from those who fulfilled the eligibility criteria set to say apparently healthy. Urinalysis and parasitological examinations (wet mount, Kato Katz and Concentration) were performed at the site of collection. About 5 ml of blood sample was collected from each study participant using plane tube in the morning from 8:00 AM to 11:00 AM. Fasting blood sugar was tested on site and recorded on a result format paper. The collected blood samples were transported by ice bag to Tigray Health Research Institute within 60 minutes for processing. Then sample was centrifuged at 2500 rpm (revolution per minute) for 5 minutes to separate serum. Serological tests for syphilis, hepatitis B and C were performed before clinical chemistry tests were done. Separated sera were stored at -40 $^0$C until analysis in a nunc tube. Biosystem 25 A (Biosystem, spain), fully automated clinical chemistry analyzer, was used for the measurement of biochemical analytes. Clinical chemistry parameters were determined by the methods/techniques described in Table 1. Selected clinical chemistry parameters such as FBS, ALP, ALT, AST, BIL. D, BIL. T, Urea, Creatinine, Total protein and Albumin values were done per the manufacturer's instructions. Stool and Urinalysis were performed onsite as part of screening apparently healthy participants.

### Quality assurance

Each activity including blood sample collection, transportation and storage were based on good laboratory practices (GLP) using standard operating procedures (SOPs) to ensure data

**Table 1. Methods used for analytes of clinical chemistry to determine the reference interval of apparently healthy Adolescents of Mekelle city, Tigrai, Ethiopia 2019.**

| Analyte | Method/technique |
|---|---|
| ALT | kinetic (IFCC without pyridoxal phosphate activation) |
| AST | kinetic (IFCC without pyridoxal phosphate activation) |
| ALP | 2-amino 2-methyl 1- propanol (AMP) |
| Cr | Jaffe Componsated |
| Urea | kinetic urease/GLDH (Glutamate dehydrogenase) |
| TP | Biuret |
| ALB | Bromocresol green |
| BIL. D and BIL. T | Diazotized sulfanilic |
| FBS | Glucose oxidase |

ALT: Alanine aminotransferase; AST: Aspartate aminotransferase; ALP: Alkaline phosphatase; Cr: Creatinine; TP: Total protein; ALB: Albumin; BIL. D: Bilirubin Direct; BIL. T: Bilirubin Total; FBS: Fasting Blood Sugar

quality. The analysis was done in Tigrai Health Research Institute (THRI) clinical chemistry laboratory which is closely supervised by Ethiopian Public Health Institute (EPHI). The equipment has been calibrated monthly by type-Autocal. In addition, two levels (normal and pathological) of internal quality control (IQC) samples were run along with the serum samples.

## Data management and analysis

Data was cleared, edited, checked for completeness manually and entered to SPSS version 23 (IBM, USA) software for analysis. Kolmogorov-Smirnov test was used to check data distribution normality. The first quartile (Q.25), the median (Q.50) and the third quartile (Q.75) were determined. Then interquartile range was calculated (IQR) from the differences of third and first quartiles (Q.75-Q.25). Data that was observed to be lower than $1.5 \times$ IQR of first quartile, or higher than $1.5 \times$ IQR of third quartile was considered as outliers and was manually deleted using the Box and Whisker statistical tool. These exclusions led to some parameter results missing hence differences in sample sizes for different parameters. Since data were not normally distributed, the Mann Whitney U test which is a non–parametric test was used to assess sex differences. The 95% RI was estimated using 2.5 th percentile for the lower reference limit and 97.5th percentile for the upper reference limit.

## Ethical considerations

This study was conducted after the study protocol was reviewed and approved by Institutional Review Board (IRB) of College of Health Sciences of Addis Ababa University. Furthermore, official permission letter was obtained from Tigrai Regional Health office. Informed written consent and assent were also obtained from each study participant and guardians before the actual data collection began. Study participants were linked to nearby health facility for any finding in urine, stool or blood.

## Results

### Demographic characteristics

A total of 344 apparently healthy adolescents (172 males and 172 females) were recruited to establish the RI for selected clinical chemistry parameters from Mekelle city, Tigrai, Ethiopia.

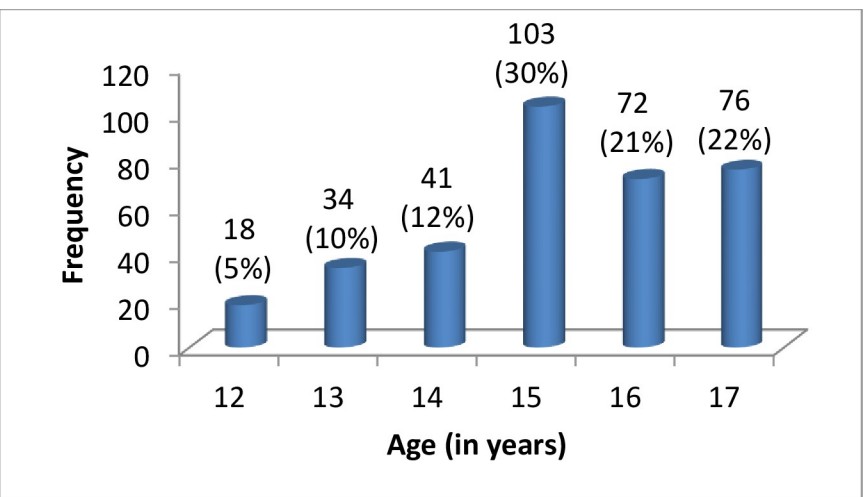

**Fig 1. Age distribution of Adolescents in Mekelle city, Tigrai Regional State, Ethiopia, 2019 (N = 344).**

The median age was 15 years [IQR 14–16 years]. The age distribution of the study participants is shown in Fig 1. A total of 45 male and 38 female study participants who had any finding in urine, stool and positive for serological tests were excluded before clinical chemistry tests analysis was performed. Of them, 25 males and 19 females had intestinal parasites. Whereas 15, 3 and 2 males and 16, 2 and 1 females were excluded based on urinalysis, serological tests and hemolysis of specimen, respectively.

### Reference intervals of clinical chemistry parameters

Table 2 summarizes the 95% RI for ten selected clinical chemistry parameters. As depicted in the table, the RIs established for adolescents in Mekelle city were FBS 65–110 mg/dl, ALP 66–456 IU/L, ALT 5–23 IU/L, AST 14.20–34.90 IU/L, BIL. D 0.03–0.53 mg/dl, BIL. T 0.10–0.81 mg/dl, Urea 8.14–24.25 mg/dl, Cr 0.37–0.91mg/dl, TP 6.09–7.85 g/dl, and Albumin 4.42–5.46 g/dl. The respective 90% confidence intervals for the lower and upper reference limits were calculated as per the recommendation of CLSI and are shown in Table 3. As shown in the table almost all lower reference limits have narrow 90% CI than the upper reference limits.

Further details of sex specific mean with 95% confidence interval, median with IQR (25th and 75th percentile) and ranges are also shown in Table 2. Comparison between males and females using the non-parametric Mann Whitney U test revealed statistically significant differences (P-value <0.005) for ALP, ALT, AST, Urea and Creatinine RIs. Except for Fasting blood sugar, Bilirubin direct and albumin all tests have higher upper limit in males than females (Table 2).

Table 4 displays comparison of the current study reference intervals with selected studies from other African countries, Western countries and company derived values. As can be seen from the table, sex aggregated reference limits are limiting. As shown in the Table, except for total protein, the other values are inconsistently varied from company derived values as well as those reported from the other countries.

## Discussion

The laboratory reference values currently employed in clinical and research institution in most of African countries including Ethiopia are referred from textbooks or from the manufacturer

**Table 2. The RIs of clinical chemistry parameters for Adolescents in Mekelle, Tigray Regional State, Ethiopia, 2019.**

| Analyte | Sex | n | Mean | Median | 95% CI of Mean | Min | Max | 25th-75th percentile | 2.5th-97.5th percentile, RI | p-value to sex |
|---|---|---|---|---|---|---|---|---|---|---|
| FBS (mg/dl) | C | 257 | 87.13 | 87.00 | 85.72, 88.53 | 60 | 118 | 79–100 | 65–110 | 0.762 |
| | M | 125 | 86.85 | 87.00 | 84.80, 88.90 | 62 | 118 | 78.50–100.50 | 64.15–108 | |
| | F | 132 | 87.39 | 87.50 | 95.44, 99.35 | 60 | 117 | 79.25–100.00 | 65.33–111.35 | |
| ALP (IU/L) | C | 259 | 201 | 166.00 | 187.59, 214.33 | 13 | 497 | 109–271 | 66–456 | <0.001* |
| | M | 123 | 271.83 | 268.20 | 252.20, 291.46 | 64 | 553 | 187–346.10 | 79.- 492 | |
| | F | 126 | 128.11 | 114.25 | 119.93, 136.28 | 13 | 285 | 98.28–145.23 | 63.56–253 | |
| ALT (IU/L) | C | 249 | 13 | 13.00 | 12.54, 13.63 | 3 | 24 | 10–16 | 5–23 | <0.001* |
| | M | 120 | 14.28 | 13.70 | 13.46, 15.11 | 3 | 25 | 11.30–17.33 | 4.54–23.69 | |
| | F | 129 | 11.74 | 11.60 | 11.11, 12.37 | 4 | 21 | 9.15–13.85 | 5.10–20.00 | |
| AST (IU/L) | C | 250 | 23 | 22.80 | 22.38, 23.67 | 10 | 40 | 19.60–26.30 | 14.20–34.90 | <0.001* |
| | M | 123 | 25.62 | 25.00 | 24.61, 26.63 | 14 | 40 | 21.80–29.30 | 15.70–39.10 | |
| | F | 127 | 20.97 | 20.80 | 20.30, 21.64 | 12 | 29 | 18.70–23.40 | 13.30–28.50 | |
| BIL. D (mg/dl) | C | 250 | 0.26 | 0.24 | 0.25, 0.28 | 0.01 | 0.60 | 0.18–0.33 | 0.03–0.53 | 0.07 |
| | M | 121 | 0.27 | 0.26 | 0.25,0.29 | 0.01 | 0.54 | 0.20–0.34 | 0.03–0.49 | |
| | F | 129 | 0.25 | 0.23 | 0.23, 0.27 | 0.01 | 0.59 | 0.16–0.32 | 0.02–0.53 | |
| BIL. T (mg/dl) | C | 248 | 0.39 | 0.37 | 0.37,0.41 | 0.02 | 0.95 | 0.26–0.49 | 0.10–0.81 | 0.059 |
| | M | 122 | 0.42 | 0.39 | 0.38,0.45 | 0.02 | 0.93 | 0.27–0.52 | 0.09–0.83 | |
| | F | 126 | 0.36 | 0.35 | 0.34, 0.39 | 0.02 | 0.81 | 0.24–0.47 | 0.10–0.72 | |
| Urea (mg/dl) | C | 257 | 15.08 | 14.65 | 14.59, 15.57 | 6 | 26 | 12.08–17.73 | 8.14–24.25 | 0.006* |
| | M | 125 | 15.87 | 15.70 | 15.14, 16.59 | 9 | 26 | 12.50–18.55 | 9.33–24.99 | |
| | F | 132 | 14.26 | 14.05 | 13.62, 14.90 | 6 | 24 | 11.65–16.80 | 7.40–23.00 | |
| Cr (mg/dl) | C | 256 | 0.66 | 0.67 | 0.64, 0.67 | 0.29 | 0.99 | 0.58–0.76 | 0.37–0.91 | 0.001* |
| | M | 122 | 0.69 | 0.69 | 0.67,0.72 | 0.37 | 1.05 | 0.59–0.79 | 0.39–0.96 | |
| | F | 134 | 0.63 | 0.64 | 0.60, 0.64 | 0.28 | 0.89 | 0.54-.72 | 0.30–0.85 | |
| T.P (g/dl) | C | 248 | 6.90 | 6.84 | 6.848, 6.96 | 5.8 | 8.40 | 6.60–7.20 | 6.09–7.85 | 0.117 |
| | M | 122 | 6.85 | 6.80 | 6.77, 6.92 | 5.8 | 8.0 | 6.51–7.14 | 5.97–7.83 | |
| | F | 126 | 6.96 | 6.85 | 6.88, 7.04 | 6.1 | 8.10 | 6.67–7.23 | 6.10–7.90 | |
| Alb (g/dl) | C | 256 | 4.88 | 4.86 | 4.85, 4.91 | 4.3 | 5.60 | 4.71–5.06 | 4.42–5.46 | 0.094 |
| | M | 124 | 4.85 | 4.83 | 4.80, 4.89 | 4.2 | 5.60 | 4.67–5.02 | 4.32–5.49 | |
| | F | 132 | 4.90 | 4.88 | 4.86, 4.95 | 4.3 | 5.60 | 4.75–5.08 | 4.42–5.46 | |

ALT: Alanine aminotransferase; ALP: Alkaline phosphatase; AST: Aspartate aminotransferase; CI: confidence interval: L: liter; mg: Milligram; dl: deciliter; RI: Reference interval; IU: International Unit; T.P: Total protein; ALB: Albumin; Cr: Creatinine; BIL. D: Direct Bilirubin; BIL. T: Total Bilirubin; C:combined; M:male;F:female; P-value < 0.05 considered as statistically significant.

of the reagent [20]. Ethiopian population depends on western derived RIs for disease diagnosis and management because of lack of locally established RIs while a number of studies showed variations between African and western population derived RIs [17–19, 21–25]. Thus, it is required to establish local RIs for adequate medical care and related health issues as well as for recruiting healthy population for clinical trials.

The upper limit of most clinical chemistry parameters among apparently healthy individuals in Africa are higher relative to Taiwan [15], Canada [13] and China [16].This difference may be due to genetic makeup, living style and altitude. The manufacturers' reference interval provided in the leaflets of the reagents used in the current study presented a single RI range for both sexes except for ALP and creatinine. However, the majority of African studies including the current study [17–20] reported separate RIs for males and females. In addition, the reference values among apparently healthy individuals of different ages are inconsistent [12]. These

**Table 3. The 90% CI for the lower and upper reference limits of clinical chemistry parameters for adolescents in Mekelle, Tigrai, North Ethiopia, 2019.**

| Analyte | unite | n | Sex | Range | 2.5th-97.5th percentile, RI | 90% CI (lower reference limit) | 90% CI (upper reference limit) |
|---|---|---|---|---|---|---|---|
| Fasting blood sugar | mg/dl | 257 | C | 60–118 | 65–110 | 63.10, 67.30 | 107.00, 116.60 |
| | | 125 | M | 62–118 | 64.15–108 | 62.10, 66.79 | 107, 118 |
| | | 132 | F | 60–117 | 65.33–111.35 | 60.40, 69.00 | 106.00, 116.50 |
| ALP | IU/L | 256 | C | 13–497 | 66–456 | 63.40, 78.30 | 433, 472 |
| | | 123 | M | 64–553 | 79.48–492.12 | 64.20, 103.70 | 457.30, 550.90 |
| | | 126 | F | 13–285 | 63.56–253.34 | 15.70, 74.30 | 221.40, 283.30 |
| ALT | IU/L | 252 | C | 3–24 | 5–23 | 4.40, 6.10 | 21.70, 24.20 |
| | | 120 | M | 3–25 | 4.54–23.69 | 2.70, 7.10 | 22.50, 25.00 |
| | | 129 | F | 4–21 | 5.10–20 | 4.4, 6.0 | 19.00, 21.30 |
| AST | mg/dl | 251 | C | 10–40 | 14.20–34.90 | 12.20, 14.90 | 33.00, 35.90 |
| | | 123 | M | 14–40 | 15.70–39.10 | 14.20, 16.90 | 35.90, 40.20 |
| | | 127 | F | 12–29 | 13.30–28.50 | 12.20, 14.90 | 27.90, 29.10 |
| BIL. D | mg/dl | 252 | C | 0.01–0.60 | 0.03–0.53 | 0.01, 0.09 | 0.49,0.58 |
| | | 121 | M | 0.01–0.54 | 0.03–0.49 | 0.01, 0.13 | 0.46,0.54 |
| | | 129 | F | 0.01–0.59 | 0.02–0.53 | 0.01, 0.10 | 0.46, 0.58 |
| BIL. T | mg/dl | 248 | C | 0.02–0.95 | 0.10–0.81 | 0.02, 0.13 | 0.72,0.84 |
| | | 122 | M | 0.02–0.93 | 0.09–0.84 | 0.02, 0.18 | 0.74, 0.93 |
| | | 126 | F | 0.02–0.81 | 0.10–0.72 | 0.02, 0.13 | 0.66, 0.81 |
| Urea | mg/dl | 258 | C | 6–26 | 8.14–24.25 | 7.40, 8.70 | 22.70, 25.00 |
| | | 125 | M | 9–26 | 9.33–24.99 | 8.90, 9.90 | 22.80, 25.50 |
| | | 132 | F | 6–24 | 7.43–23.11 | 6.20, 8.30 | 20.60, 24.30 |
| Cr | mg/dl | 253 | C | 0.29–0.99 | 0.37–0.91 | 0.32, 0.40 | 0.89, 0.93 |
| | | 122 | M | 0.37–1.05 | 0.39–0.96 | 0.38, 0.47 | 0.91, 1.05 |
| | | 134 | F | 0.28–0.89 | 0.30–0.85 | 0.28, 0.37 | 0.81, 0.89 |
| T.P | g/dl | 249 | C | 5.80–8.40 | 6.09–7.85 | 5.90, 6.20 | 7.70, 7.90 |
| | | 122 | M | 5.80–8.00 | 5.97–7.83 | 5.80, 6.20 | 7.60, 7.90 |
| | | 126 | F | 6.10–8.10 | 6.10–7.90 | 6.1, 6.3 | 7.80, 8.10 |
| ALB | g/dl | 255 | C | 4.30–5.60 | 4.42–5.46 | 4.3, 4.5 | 5.40, 5.50 |
| | | 124 | M | 4.20–5.60 | 4.32–5.49 | 4.2, 4.5 | 5.30, 5.50 |
| | | 132 | F | 4.30–5.60 | 4.42–5.46 | 4.3, 4.5 | 5.40, 5.50 |

**ALT:** Alanine aminotransferase; **ALP:** Alkaline phosphatase; **AST:** Aspartate aminotransferase; **CI:** confidence interval **:L:** liter; **mg:** Milligram; **RI:** Reference interval;
**IU: International** Unit; **dl: deciliter; g: gram;T.P:** Total protein; **ALB:** Albumin; **Cr:** Creatinine; **BIL. D:** Direct Bilirubin; **BIL.:** Total bilirubin

discrepancies may cause negative impact in the diagnosis and management of African population.

The current study's lower and upper limit results were relatively higher than the manufacture's reference value for glucose, alkaline phosphatase, albumin, Total and direct bilirubin. This difference may be due to geographical condition, genetic variation, life style and feeding culture of the study participants. On the other hand, the upper limit reference value of this study was lower when compared with the manufacturer's upper limit values for AST, ALT, Urea, creatinine and Total protein. This inconsistency may be due to feeding style, genetic variation or age differences of the population used to generate the manufacturer's derived limits.

The lower limit ALT value in the current study for both males and females was in line with the lower limit reference value generated in a study conducted in Western Kenya [17]. However, the upper limit reference value for both sexes in the current study was lower than the upper limit reference values of studies conducted in different countries including a study from

**Table 4. Comparison of clinical chemistry parameters RI of current study against manufacturer ranges and other similar studies.**

| Analyte | Sex | Current Study | Manufacturer | Taiwan [15] | Canada [13] | China [16] | Zimbabwe [6] | West Kenya [17] | Meru Kenya [18] | Taita Kenya [19] | South West Ethiopia [20] |
|---|---|---|---|---|---|---|---|---|---|---|---|
| FBS (mg/dl) | C | 65–110 | 70–100 | 60–99 | NA | NA | NA | 37.8–118.8 | NA | NA | NA |
| | M | 64.15–108 | NA | 61–98 | 75–93 | NA | 65–111.5 | NA | NA | NA | 66.80–125.80 |
| | F | 65.33–111.35 | NA | 60–99 | 75–93 | NA | 57.5–121.5 | NA | NA | NA | 59.70–117.70 |
| ALP (IU/L) | C | 66–456 | NA | NA | NA | NA | NA | NA | 61.63–114.31 | 74.70–566.60 | NA |
| | M | 79.48–492 | 0–115 | NA | 113–438 | 85–407 | 73.8–572 | NA | 61.63–114.31 | NA | 21.20–656.40 |
| | F | 63.56–253 | 0–105 | NA | 64–354 | 44–306 | 66.1–522 | NA | 58.21–114.23 | NA | 91.80–440.60 |
| ALT (IU/L) | C | 5–23 | 0–41 | 8–38 | NA | NA | NA | NA | 10.75–57.80 | 3.10–31.10 | NA |
| | M | 4.54–23.69 | NA | 8–41 | 17–50 | 7–46 | 5.8–38.8 | 4.9–42.4 | 11.18–57.20 | NA | 14.40–60.70 |
| | F | 5.10–20.0 | NA | 7–36 | 14–41 | 6–40 | 5.0–34.5 | 4.2–65.3 | 10.24–56.84 | NA | 11.00–70.50 |
| AST (IU/L) | C | 14.2–34.9 | 0–40 | NA | NA | NA | NA | NA | 9.92–54.60 | 10.00–52.01 | NA |
| | M | 15.7–39.1 | NA | NA | 18–36 | 13–38 | 14.4–45.7 | 17.0–59.2 | 10.16–54.30 | 10.00–51.10 | 12.4–58.00 |
| | F | 13.3–28.5 | NA | NA | 15–34 | 12–32 | 12.7–38.8 | 12.0–43.1 | 5.65–58.93 | 11.00–54.50 | 11.00–72.70 |
| BIL. D (mg/dL) | C | 0.03–0.53 | 0–0.2 | NA | NA | 0.06–0.43 | NA | NA | 0.02–.23 | 0.51–0.969 | NA |
| | M | 0.03–0.49 | NA | NA | NA | NA | 0.00–0.25 | NA | 0.02–0.21 | 0.13–0.94 | NA |
| | F | 0.02–0.53 | NA | NA | NA | NA | 0.007–0.2 | NA | 0.01–0.24 | 0.18–0.97 | NA |
| BIL. T (mg/dl) | C | 0.10–0.81 | 0–0.2 | NA | NA | 0.29–1.46 | NA | NA | 0.74–4.24 | 0.26–1.58 | NA |
| | M | 0.09–0.83 | NA | NA | 0.1–0.9 | NA | 0.2–1.09 | 0.33–3.66 | 0.75–4.19 | 0.22–1.54 | NA |
| | F | 0.10–0.72 | NA | NA | 0.1–0.9 | NA | 0.17–0.89 | 0.216–2.25 | 0.71–4.29 | 0.28–1.62 | NA |
| Urea (mg/dL) | C | 8.14–24.25 | 15–39 | 18.6–38.5 | NA | 9.6–39.5 | NA | 3.36–14.29 | NA | NA | NA |
| | M | 9.33–24.99 | NA | 19.2–38.5 | 8–20 | NA | 4.6–17 | NA | NA | NA | 4.60–27.20 |
| | F | 7.43–23.11 | NA | 17.9–36.6 | 8–19 | NA | 5.1–15.6 | NA | NA | NA | 3.90–48.50 |
| Cr (mg/dl) | C | 0.37–0.91 | NA | 0.4–1.1 | NA | NA | NA | NA | NA | NA | NA |
| | M | 0.39–0.96 | 0.7–1.2 | 0.4–1.2 | 0.5–0.9 | 0.37–1.05 | 0.4–1.19 | 0.56–1.17 | NA | NA | 0.30–1.90 |
| | F | 0.30–0.85 | 0.5–0.9 | 0.4–1.1 | 0.5–0.8 | 0.34–0.86 | 0.39–1.19 | 0.54–0.99 | NA | NA | 0.30–1.30 |
| T. P (g/dl) | C | 6.09–7.85 | 6.6–8.3 | NA | NA | 5.94–8.04 | NA | NA | 3.04–5.51 | 5.98–8.57 | NA |
| | M | 5.97–7.83 | NA | NA | NA | NA | 6.8–9 | NA | 3.08–5.57 | 5.88–8.4 | NA |
| | F | 6.10–7.90 | NA | NA | NA | NA | 7.1–9.06 | NA | 2.99–5.45 | 6.0–8.66 | NA |
| ALB (g/dl) | C | 4.42–5.46 | 3.5–5 | NA | NA | 3.5–5.22 | NA | NA | 2.83–4.87 | 3.56–5.3 | NA |
| | M | 4.32–5.49 | NA | NA | 4.1–5.1 | NA | 4.3–5.5 | NA | 2.79–4.89 | 3.2-8-5.3 | NA |
| | F | 4.42–5.46 | NA | NA | 4.1–5.1 | NA | 4.37–5.7 | NA | 2.86–4.85 | 3.74–5.3 | NA |

ALT: Alanine aminotransferase; ALP: Alkaline phosphatase; AST: Aspartate aminotransferase; L: liter; mg: Milligram; RI: Reference interval; IU: International Unit. NA: Not available. T.P: Total protein;ALB: Albumin; Cr: Creatinine; D.Bil: Direct Bilirubin; T.Bil: Total bilirubin;C:Combined;M:male;F:female

Southwest Ethiopia [6, 13, 15–20].This difference may be due to sample size (in china for example 2683 males and 2292 females were participated) or specimen collection time difference (In Taita Kenya, collection was performed at day time).

The Total and Direct Bilirubin reference limits in the current study were consistent with reference limit values conducted in Canada, China, Taita Kenya, Zimbabwe [6, 13, 16, 19] except the upper limit value of direct bilirubin in Zimbabwae [6] which is low. In contrast, the current upper limit provided was lower than those reported from west Kenya, Meru Kenya, and Southwest Ethiopia [17, 18, 20]. This difference may be due geographical conditions and feeding culture.

For Creatinine, both lower and upper limit reference values were comparable to several western and African studies [6, 13, 16–19] except a study from South West Ethiopia [20] where higher upper limit value was reported. In all studies females had less upper limit reference value than males. High creatinine in males compared to females may be due to the greater muscle mass in males than females [12].

## Conclusions

This study established sex specific RI for selected clinical chemistry parameters to be used for adolescents in Mekelle city. Of note, the RIs have to be validated for use by other places in the region. A statistically significant difference was noted between sexes in Alkaline Phosphatase, Alanine amino transferase, Aspartate amino transferase and Urea reference values, suggesting the need for sex specific RI for these parameters. In addition, the study observed marked difference in lower and upper limit values among the current study, the values given by the manufacturer and studies done in different countries. It underscores that every health facility set their own reference intervals.

## Acknowledgments

We thank Addis Ababa University, College of Health Science, and Department of Medical Laboratory Science for the opportunity to do this study. Our sincere thanks also go for the health extension workers who select the participants according to the selection criteria and to all study participants who kindly participated in the study. Tigrai Health Research Institute for all the support is gratefully acknowledged.

## Author Contributions

**Conceptualization:** Getachew Belay, Aster Tsegaye.

**Data curation:** Getachew Belay, Gebreyohanes Teklehaymanot, Gebreslassie Gebremariam, Kelali Kaleaye, Hagos Haileslasie, Gebremedhin Gebremichail, Brhane Tesfanchal, Getachew Kahsu, Kebede Tesfay, Lemlem Legesse, Ataklti Gebretsadik.

**Formal analysis:** Getachew Belay, Gebreslassie Gebremariam, Ataklti Gebretsadik, Aster Tsegaye.

**Investigation:** Getachew Belay, Gebreyohanes Teklehaymanot, Gebreslassie Gebremariam, Kelali Kaleaye, Hagos Haileslasie, Brhane Tesfanchal, Getachew Kahsu, Lemlem Legesse.

**Methodology:** Getachew Belay.

**Supervision:** Gebreyohanes Teklehaymanot, Gebremedhin Gebremichail, Brhane Tesfanchal, Brhane Berhe.

**Validation:** Lemlem Legesse, Ataklti Gebretsadik.

**Visualization:** Brhane Berhe, Lemlem Legesse, Aster Tsegaye.

**Writing – original draft:** Getachew Belay, Gebremedhin Gebremichail.

**Writing – review & editing:** Hagos Haileslasie, Brhane Berhe, Kebede Tesfay, Mistire Wolde, Aster Tsegaye.

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
