## [Decision Letter · Decision Letter 0]

19 Dec 2019

PONE-D-19-31034

Community based reference interval of selected clinical chemistry parameters among apparently healthy Adolescents in Tigrai, Ethiopia

PLOS ONE

Dear Mr Belay,

Thank you for submitting your manuscript to PLOS ONE. After careful consideration by two reviewers, we believe that the manuscript has merit but could be improved significantly. Therefore, we invite you to submit a revised version of the manuscript that addresses the points raised during the review process.

Specifically, as noted by Reviewer 1, more details are needed in terms of the analysis used, the socio-demographics of the participants, and the exclusion criteria used for participants. A clearer discussion section which emphasizes the main conclusions of the study would also be helpful. Please address both sets of reviewer comments and resubmit a revised version of your manuscript by Feb 02 2020 11:59PM.

To enhance the reproducibility of your results, we recommend that if applicable you deposit your laboratory protocols in protocols.io, where a protocol can be assigned its own identifier (DOI) such that it can be cited independently in the future. For instructions see: http://journals.plos.org/plosone/s/submission-guidelines#loc-laboratory-protocols

We look forward to receiving your revised manuscript.

Kind regards,

Colin Johnson, Ph.D.

Academic Editor

PLOS ONE

Journal Requirements:

2. We noticed you have some minor occurrence(s) of overlapping text with the following previous publication(s), which needs to be addressed:

https://doi.org/10.11613/BM.2016.001

https://doi.org/10.1371/journal.pone.0201782

https://doi.org/10.1016/j.cca.2019.01.001

https://doi.org/10.3109/10408363.2013.786673

https://doi.org/10.1016/j.clinbiochem.2009.06.025

https://en.wikipedia.org/wiki/Mekelle

In your revision ensure you cite all your sources (including your own works), and quote or rephrase any duplicated text outside the Methods section. Further consideration is dependent on these concerns being addressed.

4. We note you have included tables to which you do not refer in the text of your manuscript. Please ensure that you refer to Table 1 and 4 in your text; if accepted, production will need this reference to link the reader to the Tables.

5. Please amend your list of authors on the manuscript to ensure that each author is linked to an affiliation. Authors’ affiliations should reflect the institution where the work was done (if authors moved subsequently, you can also list the new affiliation stating “current affiliation:….” as necessary).

Reviewers' comments:

Reviewer's Responses to Questions

**Comments to the Author**

1. Is the manuscript technically sound, and do the data support the conclusions?

Reviewer #1: Yes

Reviewer #2: Partly

2. Has the statistical analysis been performed appropriately and rigorously? 

Reviewer #1: Yes

Reviewer #2: I Don't Know

3. Have the authors made all data underlying the findings in their manuscript fully available?

Reviewer #1: Yes

Reviewer #2: No

4. Is the manuscript presented in an intelligible fashion and written in standard English?

Reviewer #1: No

Reviewer #2: Yes

5. Review Comments to the Author

Reviewer #1: Daniel Asmelash (Questions/Comments/Suggestions)

1. The title the authors worked with is interesting and sensible for African region. However, Thorough proof reading needed

2. Justify the research gap of this study, particularly why you focus on this specific age group?

3. Representativeness of the study? Does study done only on Mekelle city will infer for the whole Tigray region? Are you inferring for the general population of Tigray region?

4. What new thing your study added from the study by Molla Abebe et. al (https://journals.plos.org/plosone/article?id=10.1371/journal.pone.0201782)

5. Socio-demographic Characteristics of the study participants lacks depth, it needs more information

6. On the Data Collection and Laboratory analysis section of the manuscript you didn’t include laboratory methods analysis for each analyte?

7. On the data analysis why, you choose Mann Whitney U test non– parametric analysis?

8. I didn’t see what type of laboratory tests was done for your exclusion criteria and the number of study participants excluded by your exclusion criteria mainly Based on laboratory findings?

9. Abreviations should be clearly mentioned on the tables

10. The Discussion needs more clarification on the possible reasons of the findings.

Reviewer #2: Thank you for the opportunity to review this manuscript titled “Community based reference interval of selected clinical chemistry parameters among apparently healthy Adolescents in Tigrai, Ethiopia”.

• There are several grammatic errors across the text.

• The authors need to reference several statements throughout the text (unfortunately there are no line numbers), e.g. first sentence in the introduction section, first sentence of paragraph 3 in the introduction section, 4th paragraph etc… several other places throughout the text.

• The authors claim in the last paragraph of the introduction section that “The results of this study, therefore, will be used as reference values in the future evidence informed practices. Patients will get better service as their result will be interpreted based on the locally established value; physicians will have better tool in their patient management process and medical laboratory professionals will have confidence especially flagged result based on RIs established elsewhere are a common challenge. Moreover, this study would serve as baseline information for further studies in the area.” Who will use these “locally established” reference intervals? How do the authors justify that their reference intervals are appropriate for the intended setting? Is a sample size of 344 enough to represent a population of 310,436 people?

• How was certainty determined for the exclusion criteria?

• How did the authors determine that the participants of the study were representative of the local population? The same criticism against the international guidelines can apply to the ‘locally established’ reference intervals.

• The tables are overwhelming. The authors need to provide a narrative description of the key messages captured in each of the tables. Additionally, based on the statistical methods described in the text and the results in the tables, the authors actually only compared local measurements with international guidelines, instead of establishing new reference intervals. There is a mismatch between the aim of the study, the methods described, and the results presented (which mostly in tables). It is unclear what the study intended to do and what was done.

6. PLOS authors have the option to publish the peer review history of their article (what does this mean?). If published, this will include your full peer review and any attached files.

Reviewer #1: Yes: Daniel Asmelash

Reviewer #2: No

---

## [Author Response · Author response to Decision Letter 0]

1 Feb 2020

Mr. Getachew Belay 

Clinical Chemistry Unit

Department of Medical Laboratory Science

College of Medicine and Health Sciences 

 Adigrat University 

P.O. Box 50

Ethiopia

Telephone +251-914480951 

getabelay65@gmail.com

Replay to Editor In-chief 

We the authors of the manuscript entitled with “Determination of community based reference interval of selected clinical chemistry parameters among apparently healthy Adolescents in Mekelle City, Tigrai North Ethiopia “had incorporated all the comments and suggestions forwarded by the editor and reviewers. The point by point response to the concerns is listed below. 

Point by point response for reviewers 

First we would like to thank and appreciate the reviewers for their critical and constructive comments. We attempted all the questions and concerns point by point as follows:

Response for reviewer 1( Daniel Asmelash)

Comment: The title the authors worked with is interesting and sensible for African region. However, thorough proof reading needed

Response: We have tried to assess all of the reviewer’s comments and revised the manuscript accordingly. 

Comment: Justify the research gap of this study, particularly why you focus on this specific age group?

Response: We were interested in this age group because most studies in Ethiopia mainly focused on adults while physiological developments during adolescence leads to changes in many analytes measured, particularly during puberty. Paragraphs 4 and 5 of the introduction section address the justifications.

Comment: Representativeness of the study? Does study done only on Mekelle city will infer for the whole Tigray region? Are you inferring for the general population of Tigray region?

Response: This comment is well taken and title is specified accordingly. Thus, the study represents only to Mekelle city adolescent population. Title is corrected as “Community based reference interval of selected clinical chemistry parameters among apparently healthy Adolescents in Mekelle City, Tigrai North Ethiopia”

Comment: What new thing your study added from the study by Molla Abebe et. al (https://journals.plos.org/plosone/article?id=10.1371/journal.pone.0201782)

Response: The current study focuses on adolescents while study participants of Molla Abebe et. al were adults. Apart from this, Clinical laboratory standard institute (CLSI) recommends every laboratory/ institute to derive its own age and sex specific reference interval. Even though there is age difference between the two studies the current upper limit of ALT, AST, Urea, Creatinine and Total bilirubin are low when we compare with the study by Molla Abebe et. al.

Comment: Socio-demographic Characteristics of the study participants lacks depth, it needs more information.

Response: Since our study participants were specific age partition, we corrected the sub-title into “Demographic characteristics” and we added a figure to show the age distribution; equal number of male and females were recruited to meet partition requirement. 

Comment: On the Data Collection and Laboratory analysis section of the manuscript you didn’t include laboratory methods analysis for each analyte?

Response: Thanks for the comment; laboratory method for each analyte is included in table form at the end of data collection and laboratory analysis.

Comment: On the data analysis why, you choose Mann Whitney U test non– parametric analysis?

Response: The data were not normally distributed when checked by statistical tool of Kolmogorov–Smirnov test. Thus, the Mann Whitney U test was used for analysis.

Comment: I didn’t see what type of laboratory tests was done for your exclusion criteria and the number of study participants excluded by your exclusion criteria mainly Based on laboratory findings?

Response: The study used three types of exclusion criteria in general; these were Physical examination, stringent questionnaire and laboratory methods. In addition to blood specimen, urine and stool specimens were collected based on good laboratory practice. Urinalysis (reagent strip and microscopy), wet mount, kato katz and concentration parasitological techniques were performed. Serological tests to syphilis, hepatitis B and hepatitis C virus were also done. Total number of samples rejected before clinical chemistry tests by laboratory methods are mentioned in the result part of the manuscript.

Comment: Abbreviations should be clearly mentioned on the tables.

Response: We attempted to mention each abbreviation on each table.

Comment: The Discussion needs more clarification on the possible reasons of the findings.

Response: We tried to give reason again on the discussion part; however, since there are limited researches conducted on this area, there is scarcity of information and we feel our contribution is worth in this regard.

Response for reviewer 2:

First we would like to thank and appreciate the reviewers for their critical and constructive comments. We attempted all the questions and concerns point by point as follows:

Comment: There are several grammatic errors across the text.

Response: English correction has been done with the help of English experts and we have corrected spelling, syntax and formatting errors in the document. 

Comment: The authors need to reference several statements throughout the text (unfortunately there are no line numbers), e.g. first sentence in the introduction section, first sentence of paragraph 3 in the introduction section, 4th paragraph etc… several other places throughout the text.

Response: Thanks; we have corrected as per the comment.

Comment: The authors claim in the last paragraph of the introduction section that “The results of this study, therefore, will be used as reference values in the future evidence informed practices. Patients will get better service as their result will be interpreted based on the locally established value; physicians will have better tool in their patient management process and medical laboratory professionals will have confidence especially flagged result based on RIs established elsewhere are a common challenge. Moreover, this study would serve as baseline information for further studies in the area.” Who will use these “locally established” reference intervals? How do the authors justify that their reference intervals are appropriate for the intended setting? Is a sample size of 344 enough to represent a population of 310,436people?

Response: The established reference interval in this study had been communicated with Mekelle city health administration in order to use the locally established values in all health facilities following same methodologies to determine each analyte. Quality was ensured at the pre analytical, analytical and post analytical phases based on standard operating procedures. In addition, the analytical part was done in Tigrai Health Research Institute, whose clinical chemistry laboratory participates in accreditation process and had three stars. This institute is mandated for all medical laboratory related issues in the region. Normal and pathological quality control serum was also run daily as well as LJ chart was interpreted based on westerngard rule. The sample size of the study was calculated based on Clinical Laboratory Standard Institute which requires a minimum of 120 study participants for each partition giving a total of 240 participants. Considering those to be excluded using the exclusion criteria, 344 participants were recruited in randomly selected sub-cities as detailed in the sampling technique section. Therefore, our study is in line with the requirement of the Clinical Laboratory Standard Institute and we believe the result can represent adolescent population of Mekelle city. Moreover, each facility (they are under the supervision of Tigrai Health Research Institute) can further validate the reference intervals on a smaller sample size.

Comment: How was certainty determined for the exclusion criteria?

Response: The study inclusion and exclusion criteria were Stringent.

The study was used three types of exclusion criteria in general; these are Physical examination, stringent questionnaire and laboratory methods. In addition to collection of blood specimen, urine and stool specimens were collected based on good laboratory practice. Urinalysis (reagent strip and microscopy), wet mount, kato katz and concentration parasitological techniques were performed. Serological tests to syphilis, hepatitis B and hepatitis C virus were also done. Study participant who had any finding was rejected from clinical chemistry test analysis.

Comment: How did the authors determine that the participants of the study were representative of the local population? The same criticism against the international guidelines can apply to the ‘locally established’ reference intervals.

Response: Thanks for this comment; also based on the recommendation of the second reviewer, we specified the title limiting the application of the RI to Mekelle city adolescents. Participants were residents who lived at east for 5 years in the study area. The study selection criteria were in line with the CLSI recommendation and the analytical part is done in Star III accredited laboratory. In addition, the laboratory is under supervision of Ethiopian Public Health Institute and participates in international accreditation process. 

Comment: The tables are overwhelming. The authors need to provide a narrative description of the key messages captured in each of the tables. Additionally, based on the statistical methods described in the text and the results in the tables, the authors actually only compared local measurements with international guidelines, instead of establishing new reference intervals. There is a mismatch between the aims of the study, the methods described, and the results presented (which mostly in tables). It is unclear what the study intended to do and what was done.

Response: Thanks for asking to clarify the finding well. Table 2 showed the newly established sex specific 2.5th and 97.5th Reference limits. The 90% confidence interval for the lower and upper limits is shown in Table 3. We, thus, provided a narrative description for clarity. On the other hand, the comparative table was included to clearly show how RIs vary between the various studies. Thank you for critically reviewing and hence enriching our manuscript.

---

## [Decision Letter · Decision Letter 1]

16 Mar 2020

Community based reference interval of selected clinical chemistry parameters among apparently healthy Adolescents in Mekelle City, Tigrai, Northern Ethiopia

PONE-D-19-31034R1

Dear Dr. Belay,

We are pleased to inform you that your manuscript has been judged scientifically suitable for publication and will be formally accepted for publication once it complies with all outstanding technical requirements.

With kind regards,

Colin Johnson, Ph.D.

Academic Editor

PLOS ONE

Additional Editor Comments (optional):

Reviewers' comments:

Reviewer's Responses to Questions

**Comments to the Author**

1. If the authors have adequately addressed your comments raised in a previous round of review and you feel that this manuscript is now acceptable for publication, you may indicate that here to bypass the “Comments to the Author” section, enter your conflict of interest statement in the “Confidential to Editor” section, and submit your "Accept" recommendation.

Reviewer #1: All comments have been addressed

2. Is the manuscript technically sound, and do the data support the conclusions?

Reviewer #1: Yes

3. Has the statistical analysis been performed appropriately and rigorously? 

Reviewer #1: Yes

4. Have the authors made all data underlying the findings in their manuscript fully available?

Reviewer #1: Yes

5. Is the manuscript presented in an intelligible fashion and written in standard English?

Reviewer #1: No

6. Review Comments to the Author

Reviewer #1: (No Response)

7. PLOS authors have the option to publish the peer review history of their article (what does this mean?). If published, this will include your full peer review and any attached files.

Reviewer #1: Yes: Daniel Asmelash Gebretensae